# Identifying Hub Genes and Metabolic Pathways in Collagen VI-Related Dystrophies: A Roadmap to Therapeutic Intervention

**DOI:** 10.3390/biom14111376

**Published:** 2024-10-29

**Authors:** Atakan Burak Ceyhan, Ali Kaynar, Ozlem Altay, Cheng Zhang, Sehime Gulsun Temel, Hasan Turkez, Adil Mardinoglu

**Affiliations:** 1Centre for Host-Microbiome Interactions, Faculty of Dentistry, Oral and Craniofacial Sciences, King’s College London, London SE1 9RT, UK; atakan.ceyhan@kcl.ac.uk (A.B.C.); ali.kaynar@kcl.ac.uk (A.K.); 2Science for Life Laboratory, KTH—Royal Institute of Technology, SE-17165 Stockholm, Sweden; oaltay@kth.se (O.A.); cheng.zhang@scilifelab.se (C.Z.); 3Department of Medical Genetics, Faculty of Medicine, Bursa Uludag University, Bursa 16059, Turkey; sehime@uludag.edu.tr; 4Department of Translational Medicine, Institute of Health Science, Bursa Uludag University, Bursa 16059, Turkey; 5Department of Histology and Embryology, Faculty of Medicine, Bursa Uludag University, Bursa 16059, Turkey; 6Department of Medical Biology, Faculty of Medicine, Atatürk University, Erzurum 25030, Turkey; hturkez@atauni.edu.tr

**Keywords:** systems biology, collagen VI-related dystrophies, network analysis, drug repurposing, genome-scale metabolic models

## Abstract

Collagen VI-related dystrophies (COL6RD) are a group of rare muscle disorders caused by mutations in specific genes responsible for type VI collagen production. It affects muscles, joints, and connective tissues, leading to weakness, joint problems, and structural issues. Currently, there is no effective treatment for COL6RD; its management typically addresses symptoms and complications. Therefore, it is essential to decipher the disease’s molecular mechanisms, identify drug targets, and develop effective treatment strategies to treat COL6RD. In this study, we employed differential gene expression analysis, weighted gene co-expression network analysis, and genome-scale metabolic modeling to investigate gene expression patterns in COL6RD patients, uncovering key genes, significant metabolites, and disease-related pathophysiological pathways. First, we performed differential gene expression and weighted gene co-expression network analyses, which led to the identification of 12 genes (*CHCHD10*, *MRPS24*, *TRIP10*, *RNF123*, *MRPS15*, *NDUFB4*, *COX10*, *FUNDC2*, *MDH2*, *RPL3L*, *NDUFB11*, *PARVB*) as potential hub genes involved in the disease. Second, we utilized a drug repurposing strategy to identify pharmaceutical candidates that could potentially modulate these genes and be effective in the treatment. Next, we utilized context-specific genome-scale metabolic models to compare metabolic variations between healthy individuals and COL6RD patients. Finally, we conducted reporter metabolite analysis to identify reporter metabolites (e.g., phosphatidates, nicotinate ribonucleotide, ubiquinol, ferricytochrome C). In summary, our analysis revealed critical genes and pathways associated with COL6RD and identified potential targets, reporter metabolites, and candidate drugs for therapeutic interventions.

## 1. Introduction

Muscular dystrophies are a group of congenital disorders resulting from gene mutations that play an essential role in muscle function and structure. Their clinical symptoms are progressive, characterized by muscle weakness and disability [1,2]. There are various types of muscular dystrophy, each with distinct onset times, severity levels, and life expectancies. COL6RD represents a subgroup of muscular dystrophies, extending from the less severe Bethlem muscular dystrophy to the more severe Ullrich muscular dystrophy [3]. These conditions result from genetic mutations affecting the genes responsible for producing the three primary α-chains of collagen type VI: *COL6A1*, *COL6A2*, and *COL6A3* [4]. COL6RD is associated with multiple metabolic dysfunctions, especially those impacting mitochondria and cellular energy production [4,5]. Unfortunately, there is not an FDA-approved drug specifically developed for COL6RD [4]. The treatment primarily aims at managing symptoms and complications, including physical therapy, respiratory support, orthopedic care, nutritional assistance, and cardiac evaluations [4].

Systems biology is an interdisciplinary field that explores the interactions of biological components as an integrated system, rather than analyzing them individually. The development of high-throughput technologies and enhanced computational capabilities has given rise to systems biology as a novel field of research [6]. Its methodologies are employed to comprehend complex diseases, uncover molecular mechanisms, and accelerate biomarker and drug discovery [7,8,9,10,11,12,13]. It combines experiments and computational analysis to explore how biological components operate, employing diverse methods such as bioinformatic tools, network modeling, and computational modeling [14,15]. These approaches are substantial for gaining a comprehensive understanding of biology. For example, weighted gene co-expression network analysis (WGCNA) has been effectively used in previous studies to build gene networks in diverse diseases, identifying hub genes as potential biomarkers and therapeutic targets [16,17]. Furthermore, genome-scale metabolic models (GEMs) have been applied in various contexts, they offer detailed insight into the connections between metabolites, genes, and reactions in the organisms by consolidating current literature knowledge and omics data [18,19,20].

Drug repurposing, exploring new uses for existing medications, is gaining popularity due to the high costs and low success rates of new drug development [21]. This method utilizes approved drugs or well-characterized drug candidates, potentially cutting down costs and development time [22]. Approaches include experimental and computational methods such as analyzing disease pathways, drug interactions, multi-omics data integration, and clinical data mining [23]. As a successful example, combined metabolic activators (CMAs) have been repurposed for treating mitochondrial dysfunction-associated diseases, including neurodegenerative diseases, nonalcoholic fatty liver disease, and COVID-19; their effects have been demonstrated in preclinical and clinical studies [24,25,26,27,28,29]. Previously, anti-TGFβ antibodies [30], SB431542 [30], losartan [30], and cyclosporin A [31,32] have been suggested for treating COL6RD based on previous studies, but further research and clinical trials are necessary to assess their safety and effectiveness. Within this context, employing a systems biology approach could assist in repurposing current medications and innovating new treatment options for COL6RD.

In this study, we employed systems biology methodologies to understand the molecular pathophysiology of COL6RD, decipher the underlying molecular mechanisms of the disease, and identify potential drug targets that could aid in developing more effective treatment options. First, we performed differential expression analysis and gene set enrichment analyses and identified the upregulated and downregulated genes and their associated pathways along two conditions. Secondly, we generated a weighted gene co-expression network to reveal the genes that work collaboratively in healthy and diseased states. It enabled the identification of the hub genes that have important roles in the occurrence and progression of the disease. Thirdly, we applied a pathway-based drug repurposing method to discover potent drug candidates by calculating the correlation between gene perturbation profiles and functional enrichment results of the target genes. Finally, we reconstructed context-specific genome-scale metabolic models for both COL6RD patients and the control group. This allowed us to identify reporter metabolites and present the global changes at the metabolite level. The general outflow of the study is represented in Figure 1.

## 2. Materials and Methods

### 2.1. Transcriptomics Data and Pre-Processing

In this study, we employed previously generated global RNA-sequencing (transcriptomics) data of muscle tissues obtained from 22 patients with COL6RD and 14 healthy controls [33]. First, raw fastq files were obtained from the Gene Expression Omnibus database (accession number: GSE103608, access date: 8 April 2024) with the aid of “prefetch” and “fasterq-dump” functions of the SRA Toolkit [34]. Additionally, we downloaded the latest human reference genome (GRCh38.p14) from the Ensembl database [35]. First, an index file was created using the “index” function. Then, transcript abundances were calculated using the “quant” function of Kallisto (version 0.46.1) [36]. The expression level of each gene was calculated by summing the values of all its transcripts. We selected only protein-coding genes annotated in the BioMart data mining tool [37] because a core objective of our study is to identify protein targets and modulate their activity with drugs. The metadata of the samples (age, sex, clinical phenotype, etc.) can be found in the Appendix A.

### 2.2. Differential Gene Expression and Gene Set Enrichment Analyses

We used the “DESeq2” package in R to conduct the differential gene expression analysis [38]. Initially, a matrix was generated by correlating clinical data with raw count numbers from COL6RD patients and the control group. Later, we eliminated genes with total counts below 10 across all samples to achieve more reliable outcomes and minimize noise. Subsequently, the analysis was run by choosing the healthy control group as the baseline condition to detect differentially expressed genes (DEGs). DEGs with an adjusted *p*-value smaller than 10^−5^ were considered statistically significant for our downstream analysis. DEGs with a positive log2FoldChange value were labeled as upregulated, while those with a negative log2FoldChange value were labeled as downregulated. We utilized the “ggplot” package in R to visualize the results of the differential gene expression analysis [39].

For gene set enrichment analyses, the “enrichGo” function of the clusterProfiler package in R was employed to identify significantly enriched genes utilizing the hypergeometric distribution for biological process pathways [40]. A *p*-value < 0.05 was employed as a cut-off to pinpoint significantly enriched pathways. The “org.Hs.eg.db” package was used for human gene annotation and only the biological process (BP) ontology parameter was selected for the analyses [41]. The “enrichplot” package was used for illustration of the gene set enrichment outcomes.

### 2.3. Weighted Gene Co-Expression Network Analysis

Weighted gene co-expression network analysis was used to identify co-expression patterns among genes and their relationship with certain conditions. To build co-expression networks, we employed the WGCNA package in R [42]. As a first step, we undertook quality control utilizing the “goodSamplesGenes” function that helped to exclude genes with many incomplete records. By this means, we ensured that the analysis was conducted using reliable data. Later, a cluster tree map was created to find and exclude outlier samples. Next, genes with low counts (fewer than 15 in 75% of samples) were removed to ensure that the remaining genes had sufficient expression levels in most samples. Thus, we enabled the detection of more robust co-expression patterns.

“DESeq2” was employed for variance stabilization and normalization of the count data, in order to maintain comparable expression levels across samples. Next, correlation coefficients were calculated between every pair of genes to generate a gene expression similarity matrix, representing the strength of their co-expression relationships. The package’s “pickSoftThreshold” function was used to select the optimal soft threshold value, aiming to achieve a scale-free network that has a minimum scale independence of 0.8. Subsequently, the adjacency matrix was derived from the gene expression similarity matrix. Then, it was converted into a topological overlap matrix that captures both direct and indirect relationships between genes using hierarchical clustering. Dynamic cut tree methods were used to detect gene co-expression network modules (with the “minCoreKME” parameter set to 0.5 and the “mergeCutHeight” parameter set to 0.25). The topological overlap matrix (TOM) was designated as “signed” to identify gene modules that were positively correlated, indicating they have similar biological functions.

After building modules, we calculated their eigengenes, which represent the dominant expression pattern shared by genes within the modules. Following, the correlation between module eigengenes and disease state was measured with the help of the Pearson correlation coefficient [43]. Modules with a *p*-value below 0.05 and a positive correlation with COL6RD were termed upregulated modules, while those with a negative correlation were termed downregulated modules. Last, we employed intramodular analysis to pinpoint hub genes by measuring module membership parameters along with *p*-values.

### 2.4. Defining Gene Targets

Genes in each module were ranked in decreasing order according to their module membership scores. Subsequently, the top 1% of genes from every module were identified and extracted. Initially, we examined the overlap between upregulated and downregulated DEGs and the top 1% of upregulated and downregulated modules. Next, a more detailed investigation of these genes within the Human Protein Atlas Database was conducted [44]. We selected genes with certain RNA expression levels in muscle tissue (tissue-enriched or tissue-enhanced) as our targets. Since COL6RD primarily affects muscles, we hypothesized that targeting muscle-specific genes would reduce systemic side effects and increase the therapy’s success rate when we manipulated these genes with drugs.

### 2.5. Pathway-Based Drug Repositioning

In this study, we used a web-based tool known as Gene2drug, which leverages gene expression data sourced from the Connectivity Map [45,46]. These data were transformed into pathway expression profiles. Then, they were sorted based on the *p*-value of the Kolmogorov–Smirnov statistic. Subsequently, when a collection of pathways was provided, the Kolmogorov–Smirnov statistic was reapplied to identify drugs that consistently induce upregulation or downregulation across most pathways within the set.

We repeated gene set enrichment analyses on our specified gene targets from the previous step. Following this, we integrated biological process gene ontology results into the Gene2drug tool and downloaded all the findings in an Excel file. We selected only drugs with positive enrichment scores that indicate pathway upregulation and *p*-values below 0.05. We excluded antineoplastic agents due to their cytotoxic effects.

### 2.6. Genome-Scale Metabolic Modeling and Reporter Metabolite Analysis

We separately computed the average transcripts per million (TPM) values for each gene in the COL6RD patients and the healthy control group. Genes with an average TPM of at least 1 were removed to reduce noise from low-expression genes in the cohort. Subsequently, we used these values as the starting point for creating specialized GEMs with the help of a task-driven integrative network inference algorithm called tINIT using Human-GEM v1.18 as the reference model [8,47]. This algorithm automated the creation of GEMs by incorporating omics data. Modeling was conducted in MATLAB R2023a (MathWorks, Natick, MA, USA) with the RAVEN v2.8.6 toolbox for constructing the models and the Gurobi 11.0 (Gurobi Optimization LLC, Beaverton, OR, USA) solver for optimization [48]. Before advancing in our pipeline, we assessed the models to determine if they could successfully execute 57 predefined metabolic tasks, referred to as essential tasks in the reference model [8]. These tasks encompassed vital cellular functions, e.g., protein construction, ATP production, and nucleotide synthesis. Five GEMs were built to represent healthy controls, COL6RD patients with 3 different histological states, and a common COL6RD model to describe all patients’ average expression profiles.

The reporter metabolites analysis (RMA) was carried out employing the “reporterMetabolites” function within the RAVEN Toolbox. This worked through results integration from the differential gene expression analysis into the common COL6RD model. The analysis pinpointed metabolites that were influenced by alterations in metabolic networks and gene expression levels in COL6RD patients. Before running the RMA, 20 currency metabolites (“H_2_O”, “CO_2_”, “O_2_”, “H+”, “HCO3−”, “Na+”, “CoA”, “Pi”, “PPi”, “AMP”, “ADP”, “ATP”, “NAD+”, “NADH”, “NADP+”, “NADPH”, “PAP”, “PAPS”, “FAD”, and “FADH2”) were excluded from the model because their widespread involvement in reactions can obscure specific changes, add noise, and reduce clarity. Their removal highlighted relevant pathway-specific changes and improved statistical power. Our primary focus was on metabolites exhibiting lower *p*-values than 0.05, and RMA was conducted separately for those linked to downregulated and upregulated genes.

To investigate diversities between common COL6RD and the healthy GEMs, the hamming distance was computed for each couple of models, based on the count of distinct reactions present exclusively in one of the models. Additionally, the findings were visually represented through a cluster gram. Furthermore, differences in subsystem coverage were demonstrated for subsystems showing at least a 25% variation in one or more GEMs, emphasizing significant alterations in metabolic processes under varying conditions. Also, we created a dot plot to illustrate the diversity in metabolic networks among various models. The diversity was quantified using the “compareMultipleModels” function from the RAVEN Toolbox. For this analysis, we utilized an extensive metabolic task file containing 256 tasks, named “metabolicTasks_Full.xlsx”, which was in the Human-GEM repository.

## 3. Results

### 3.1. Analysis of Transcriptomic Data from COL6RD Patients Reveals Distinctive Gene Signatures

We performed the differential gene expression analysis for transcriptomics data of COL6RD patients versus healthy controls to identify the changes in gene signatures at the disease state. The cohort included samples from 22 muscular dystrophy patients and 14 healthy individuals. Consequently, we detected 2528 DEGs within the cohort (with a *p*-adjusted value < 10^−5^), comprising 1679 upregulated genes and 849 downregulated genes in COL6RD samples relative to the control samples (Figure 2A).

Functional enrichment analysis revealed that upregulated DEGs were significantly enriched in extracellular matrix and structure organization, cell-substrate adhesion, bone development, collagen fibril organization, axonogenesis, cell–cell adhesion via plasma membrane adhesion molecules, and ossification (Figure 2B). On the other hand, downregulated DEGs were predominantly enriched in aerobic respiration and its associated processes, including mitochondrial respiratory chain complex assembly, oxidative phosphorylation, proton motive force-driven mitochondrial ATP synthesis, ATP metabolic process, and mitochondrial gene expression (Figure 2C). Likewise, the initial research paper emphasized the increased activity of extracellular matrix and structural organization pathways while noting a decrease in mitochondrial pathways [33]. However, there were notable differences. For instance, we identified 2528 DEGs, whereas the original study reported only 248. Additionally, our gene ontology analysis revealed oxidative phosphorylation and mitochondrial energy metabolism as significant pathways for downregulated DEGs, contrasting with the original study’s emphasis on fiber contraction and metabolic processes.

### 3.2. Weighted Gene Co-Expression Network Analysis Reveals the Gene Modules for COL6RD

We generated a gene co-expression network matrix using the raw count from the RNA-seq data to investigate functional gene modules. During the initial quality control phase, 1361 genes were eliminated because of either having excess missing values or zero variance. Later, the hierarchical clustering method detected two outlier samples (SRR6015106, SRR6015079) and omitted them from the analysis (Figure 3A). Following that, 7886 genes that had less than 15 counts in 75% of samples were eliminated for the downstream pipeline. After normalizing the counts using the Deseq2 package, the soft threshold power of 9 was selected by reason of yielding high R2 results and lower mean connectivity (Figure 3B). Subsequently, an adjacency matrix was constructed, and the topological overlap measure along with dynamic tree cutting using a cut-off value of 0.25 was applied to generate clusters (Figure 3C). This process resulted in the identification of 12 modules, with module sizes varying from 24 to 4872.

Following the computation of module eigengenes for each module, the Pearson correlation coefficient was determined between module eigengenes and the disease state. Among the 12 modules tested, 9 displayed statistically significant results (*p* < 0.05). Within these modules, six were classified as upregulated (yellow, green-yellow, magenta, green, red, brown), while three were categorized as downregulated (blue, pink, turquoise) (Figure 3D). When examining gene set enrichment analyses of upregulated modules, various distinct patterns emerge. For instance, the yellow module was enriched in processes such as ncRNA processing and endosomal transport, while the green-yellow module shows enrichment in the planar cell polarity pathway and epithelial tube morphogenesis. The green module exhibits enrichment in pathways related to immune response activation via cell surface receptors and myeloid leukocyte activation. In contrast, the red module demonstrates enrichment in lipid catabolism and alcohol metabolism, and the brown module was enriched in extracellular matrix organization. Interestingly, the magenta module did not show enrichment in any specific biological pathways. Similarly, downregulated modules also displayed unique enrichment patterns. For example, the blue module was enriched in aerobic and cellular respiration as well as oxidative phosphorylation. The pink module exhibits enrichment in cytoplasmic translation and ribosome biogenesis, while the turquoise module shows enrichment in RNA splicing and ribosome biogenesis. The results of functional enrichment analysis for all gene modules are presented in the Appendix A.

### 3.3. Identification of the Hub Genes as Drug Targets

Out of the co-expression network modules, we selected only the brown and blue modules for our downstream analysis as they have both the lowest *p*-value and the highest correlation score with the disease state. Upon examining the intersection of the top 1% of the brown module, as determined by the module membership parameter and the upregulated DEGs, we discovered 22 common genes (Figure 4A). However, second step control from the Human Protein Atlas dataset revealed that none of these genes exhibited muscle tissue-specific expression RNA levels. Therefore, we did not select any of them as targets. On the other hand, when comparing the top 1% of the blue module with the downregulated DEGs, we identified 24 common genes. Among these 24 genes, 12 displayed muscle tissue-specific RNA expression levels (Figure 4B). As a result, we selected these genes as our targets: *CHCHD10*, *MRPS24*, *TRIP10*, *RNF123*, *MRPS15*, *NDUFB4*, *COX10*, *FUNDC2*, *MDH2*, *RPL3L*, *NDUFB11,* and *PARVB* (Table 1).

To provide a more comprehensive overview of COL6RD’s gene expression profile, we also examined *COL6A1*, *COL6A2*, and *COL6A3* gene co-expression patterns using the iNetmodels database [61,62]. We observed several positive correlations between *COL6A1* and *COL6A2*, but no correlations were observed with COL6A3. This indicated that COL6A1 and *COL6A2* had a stronger correlation with each other compared to COL6A3. This variation might result from *COL6A3* being regulated diversely and having additional biological functions compared to the other two alpha chains [63]. Furthermore, when examining all the connection nodes among *COL6A1*, *COL6A2*, and *COL6A3*, we found that 46 nodes exhibited upregulation in our dataset, whereas 9 nodes showed no differential expression in our study. Figure 4D presents this co-expression comparison map and the correlations between the nodes.

### 3.4. Drug Repositioning to Identify Potential Drug Candidates

We utilized gene ontology terms representing biological pathways obtained from gene set enrichment analysis of 12 target genes as input for the Gene2drug database (Figure 4E). This yielded 222 drugs (*p* < 0.05), among which 64 drugs were found to upregulate our target pathways (with positive enrichment scores), while 158 drugs downregulated them (with negative enrichment scores). Given that all our gene targets for COL6RD were downregulated, we aimed to upregulate them to treat the disease and restore cell status closer to the healthy state. Consequently, we focused on drugs with positive enrichment scores. Next, we selected the top 10 drugs based on the lowest *p*-values, excluding antineoplastic agents due to their cytotoxic effects. These drugs include apigenin, flunarizine, deferoxamine, luteolin, verteporfin, ursodeoxycholic acid, ioxaglic acid, risperidone, fipexide, and naftifine. More detailed information about these drugs can be found in Table 2.

### 3.5. Comparison Between the Context-Specific Genome-Scale Metabolic Models

To comprehend the metabolic context of the disease, we constructed four GEMs for COL6RD and one model for a healthy state based on their transcriptomic profiles using tINIT. The COL6RD models include separate representations for three histology grades, denoted as COL6RD-1, COL6RD-2, and COL6RD-3. Additionally, a common COL6RD model was developed by averaging the TPM values across all muscular dystrophy patients at all three stages, providing a general view of the disease. Following this, we carried out an analytical evaluation using methodologies based on GEMs. The GEMs for COL6RD-1, COL6RD-2, COL6RD-3, and COL6RD-Common varied in their numbers of genes, reactions, and metabolites. Specifically, COL6RD-1 comprised 1877 genes, 6810 reactions, and 5621 metabolites, while COL6RD-2 included 1899 genes, 6576 reactions, and 5258 metabolites. Similarly, COL6RD-3 consisted of 1907 genes, 6697 reactions, and 5375 metabolites, and COL6RD-Common encompassed 1909 genes, 7038 reactions, and 5880 metabolites. In contrast, the healthy model consisted of 1740 genes, 6409 reactions, and 5280 metabolites.

Upon inspection of the Hamming distance between the GEMs, it was observed that the healthy and COL6RD-Common models exhibit the most distinct pattern, as expected. Furthermore, COL6RD-3 and COL6RD-2 displayed the highest similarity, with a Hamming distance score of 0.96. This was followed by the COL6RD-1 and COL6RD-Common models (Figure 5A). When looking at subsystem coverage between models, we saw upregulation in peptide metabolism in the healthy model compared to COL6RD-Common. In contrast, the healthy model had downregulation in vitamin B2 metabolism, beta-oxidation of fatty acids, bile acid metabolism, and phenylalanine metabolism compared to the disease model (Figure 5B). DPA, DHA, 12,15,18,21-tetracosatetraenoic acid and adrenic acid de novo synthesis were represented in the healthy model but were not represented in the COL6RD-Common model. On the other hand, various tasks were represented in the COL6RD-Common model and not in the healthy model such as ATP salvage from hypoxanthine, lauric acid myristic acid de novo synthesis, etc. (Figure 5C).

### 3.6. Revealing the Differences by Performing Reporter Metabolite Analysis

RMA reveals important key metabolites that are associated with significant changes in gene expression levels. In this study, we identified 224 upregulated and 73 downregulated reporter metabolites for COL6RD (*p* < 0.05) (Figure 6A,B). Namely, the top five upregulated metabolites were nicotinate ribonucleotide, phosphatidate-LD-PC pool, phosphatidate-LD-PE pool, phosphatidate-LD-PS pool, and (7Z)-octadecenoyl-CoA, while ubiquinol, ubiquinone, ferricytochrome C, ferrocytochrome C, and 3-phospho-D-glycerate were the top five downregulated metabolites, based on ascending *p*-values (Table 3). We observed that these metabolites were mostly associated with mitochondrial dysfunction [74,75,76].

## 4. Discussion

To reveal the molecular mechanisms involved in COL6RD, we employed transcriptomic data from a prior cohort study. We conducted differential expression and gene ontology analysis to understand molecular patterns of the disease. Although these two analyses were conducted in the original study as well, our study provided different results due to using the latest reference genome (GRCh38.p14) and different bioinformatic tools (kallisto, DESeq2, clusterProfiler, etc.). In addition, our research has extended the original findings through further downstream analyses. For example, using gene co-expression analysis, we found gene modules and hub genes linked to COL6RD. We also identified candidate drugs suitable for therapeutic intervention by employing a pathway-based drug repurposing strategy. Also, genome-scale metabolic models for both diseased and healthy conditions were reconstructed to compare the global metabolic changes in response to COL6RD. Lastly, we identified important changes at the metabolite level by utilizing reporter metabolite analysis. To sum up, our analysis uncovers insights into the molecular pathophysiology of COL6RD, identifying potential drug candidates and reporter metabolites to advance its diagnosis, screening, and treatment strategies.

In this study, we observed downregulation in mitochondria and aerobic respiration processes in COL6RD. Similarly, previous research has shown that in mouse models and muscle cells from patients with COL6RD, mitochondria exhibit structural and functional anomalies, including enlarged size, reduced matrix density, distorted cristae, and impaired respiratory capacity [85,86,87]. While these conditions are associated with mitochondrial dysfunction and impaired oxidative phosphorylation, they lead to reduced ATP production and increased oxidative stress in muscle cells [85,88]. The combination of mitochondrial dysfunction and oxidative stress leads to muscle fiber apoptosis, which worsens muscle damage and weakness in muscular dystrophies, making them key pathogenic mechanisms.

Examination of the iNetModels database indicated that 10 out of 12 of our target genes were interconnected (Figure 4C). The proteins were mainly located in mitochondria and play roles in oxidative phosphorylation. These genes were shown as drug targets for many muscular diseases by previous researchers. For instance, *COX10* encodes a key component of cytochrome c oxidase, with mutations linked to congenital muscular dystrophy and mitochondrial myopathies [89,90,91]. Another example is *MDH2* (malate dehydrogenase 2), a mitochondrial enzyme identified as a promising prognostic biomarker for Duchenne muscular dystrophy [92]. Serum levels of *MDH2* significantly decrease with age in Duchenne muscular dystrophy patients, correlating with disease progression [93]. Likewise, prior studies aimed to boost mitochondrial activity in COL6RD patients using cyclosporine A and its non-immunosuppressive derivatives [31,86]. The treatment improved mitochondrial function, muscle regeneration, and reduced apoptosis, supporting these compounds as potential therapies for mitochondrial dysfunction in collagen VI-related dystrophies. In this context, CMAs can also be used to activate the mitochondrial metabolism in the muscle tissue of COL6RD patients since their effect in activating mitochondria has been demonstrated previously [24,25,26,27,28,29].

Our repurposed drugs, apigenin and luteolin, are flavonoids that occur naturally in fruits and vegetables. They exhibit anti-inflammatory, antioxidant, and anticancer characteristics, offering potential advantages for diverse cancers and other oxidative-associated ailments [94,95,96]. They are recommended for muscular dystrophies and muscle atrophy due to their ability to enhance muscle growth and myogenic differentiation, and reduce oxidative stress [97,98,99,100,101]. Another medication recommended through our pathway-based drug repurposing is deferoxamine, which functions as a chelating agent utilized in the treatment of iron or aluminum toxicity [102]. Previous studies propose its potential use in treating Duchenne muscular dystrophy by preventing oxidative damage through the removal of surplus iron, potentially aiding in the reversal of muscle damage associated with the condition [103,104]. While these proposed medications have not been specifically tested for COL6RD, their potential effectiveness is anticipated based on similarities in disease mechanisms with other muscular dystrophies.

In this research, we created context-specific GEMs to represent both healthy controls and COL6RD patients in three histologic states, comparing them to identify metabolic differences. We found increased lipid metabolism and decreased energy metabolism in COL6RD patients. Later, reporter metabolite analysis showed elevated lipid-associated metabolites, e.g., phosphatidates, and reduced energy-related metabolites such as ubiquinol and ferricytochrome C. These findings align with existing literature knowledge. For instance, it is known that in many muscular dystrophies, adipose tissue takes over muscle tissue, a process known as fatty infiltration [105,106,107]. Likewise, previous research indicates that COL6RD is linked to obesity and causes a rise in overall adipose tissue, emphasizing the need to consider this in patient management [108]. Additionally, several prior studies have observed a decrease in energy metabolism in COL6RD patients [85,86,88]. Since reporter metabolites can be detected in plasma and urine and hold significance as biomarkers [75], we hypothesized that phosphatidate, ubiquinol, ferricytochrome C, and related metabolites might be candidate biomarkers for COL6RD. Nonetheless, further metabolomic and clinical studies are necessary to confirm these results.

Our study faces three main limitations. First, the small sample size (*n* = 36) restricts the generalizability of our findings, despite the rarity of the disease. Second, the cohort consists only of cross-sectional samples, which prevents us from observing changes over time. Lastly, our study is based solely on computational analysis without biochemical or molecular validation. To address these issues, we plan to collect more samples from COL6RD patients and validate our findings using Western blot experiments and metabolomic analysis. Although our study makes modest contributions to the understanding of COL6RD, it provides valuable insights and lays the groundwork for further research.

## 5. Conclusions

In conclusion, our study on COL6RD using systems biology has offered important insights into the disease’s pathophysiology. By integrating differential expression analysis with WGCNA, we pinpointed hub genes and identified 12 potential drug targets. Then, we applied the pathway-based drug repositioning strategy to discover pharmacotherapy candidates aimed at modulating these genes. Additionally, we explored metabolic differences between healthy individuals and COL6RD patients by reconstructing and comparing context-specific genome-scale metabolic models. This was followed by reporter metabolite analysis, which helped identify key metabolites that may be valuable for early detection and monitoring. Briefly, our study identified hub genes, drug targets, key metabolites and pathways linked to COL6RD, and promising drug candidates for improving patient care. It provides a foundation for future metabolomic and clinical research to validate these findings.

## Figures and Tables

**Figure 1 biomolecules-14-01376-f001:**
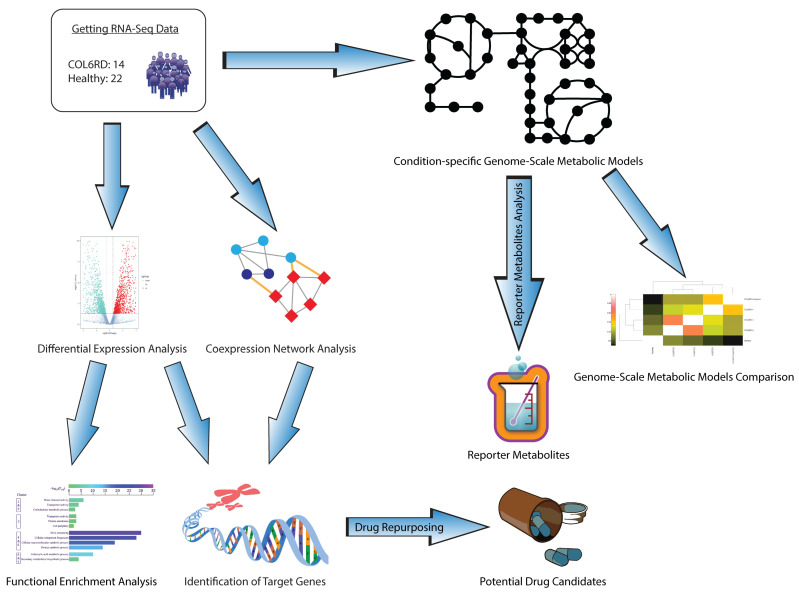
A blueprint illustrating a systematic approach for our research. Initially, we analyzed RNA sequencing data from a prior cohort to pinpoint gene targets. This involved conducting differential expression, functional enrichment, and gene co-expression network analysis, then integrating their findings. Subsequently, we employed a pathway-based method for drug repurposing to identify drugs that can activate these genes in COL6RD. Additionally, we developed condition-specific GEMs to highlight metabolic distinctions between COL6RD patients and healthy controls. Finally, we utilized reporter metabolite analysis to propose potential biomarkers for COL6RD.

**Figure 2 biomolecules-14-01376-f002:**
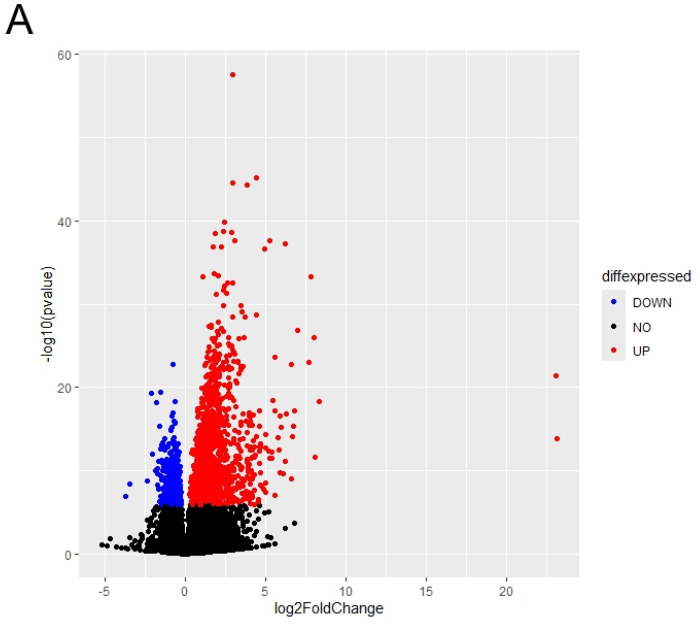
(**A**) The distribution of differentially expressed genes is depicted. Genes exhibiting significantly upregulated expressions (FDR < 10^−5^) are highlighted in red, while those with significantly downregulated expressions (FDR < 10^−5^) are depicted in blue. Genes with FDR scores exceeding 10^−5^ are illustrated in black. (**B**,**C**) The top 20 enriched pathways of upregulated DEGs are presented in (**B**), while the top 20 enriched pathways of downregulated DEGs are displayed in (**C**), based on the *p*-adjusted score. In this representation, a smaller *p*-adjusted value is denoted by the color red, while blue indicates a higher value. The size of the dots reflects the number of genes associated with a specific pathway.

**Figure 3 biomolecules-14-01376-f003:**
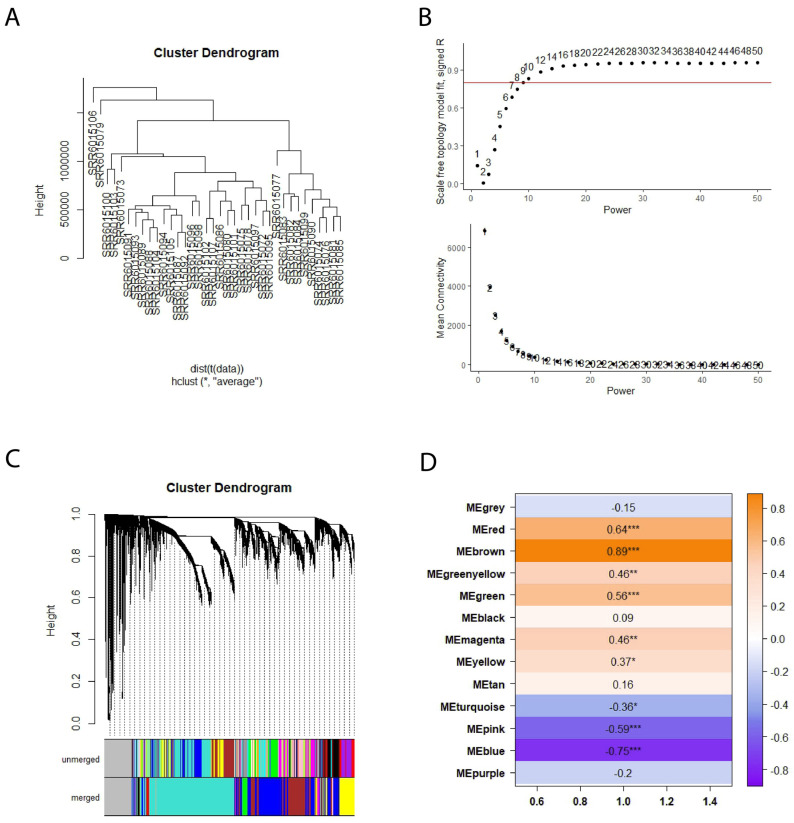
(**A**) The dendrogram illustrating the hierarchical clustering of the COL6RD cohort’s samples was generated, but two outliers (SRR6015106 and SRR6015079) were excluded. (**B**) Graphs were created to depict the mean connectivity and fit of the scale-free topology model, with the y-intercept set at 0.8. Based on a high R2 value and reduced mean connectivity, a soft threshold power of 9 was chosen. (**C**) Profiles of cluster dendrograms and module detection are displayed, with module colors shown before and after merging below the dendrogram. (**D**) The heatmap shows Pearson correlations between module eigengenes and COL6RD disease state, with positive correlations in brown and negative in purple. Asterisks indicate significance levels: a single asterisk (*) denotes a *p*-value below 0.05, a double asterisk (**) denotes a *p*-value below 0.01, and a triple asterisk (***) denotes a *p*-value below 0.001. Genes that are not part of any module are shown in grey.

**Figure 4 biomolecules-14-01376-f004:**
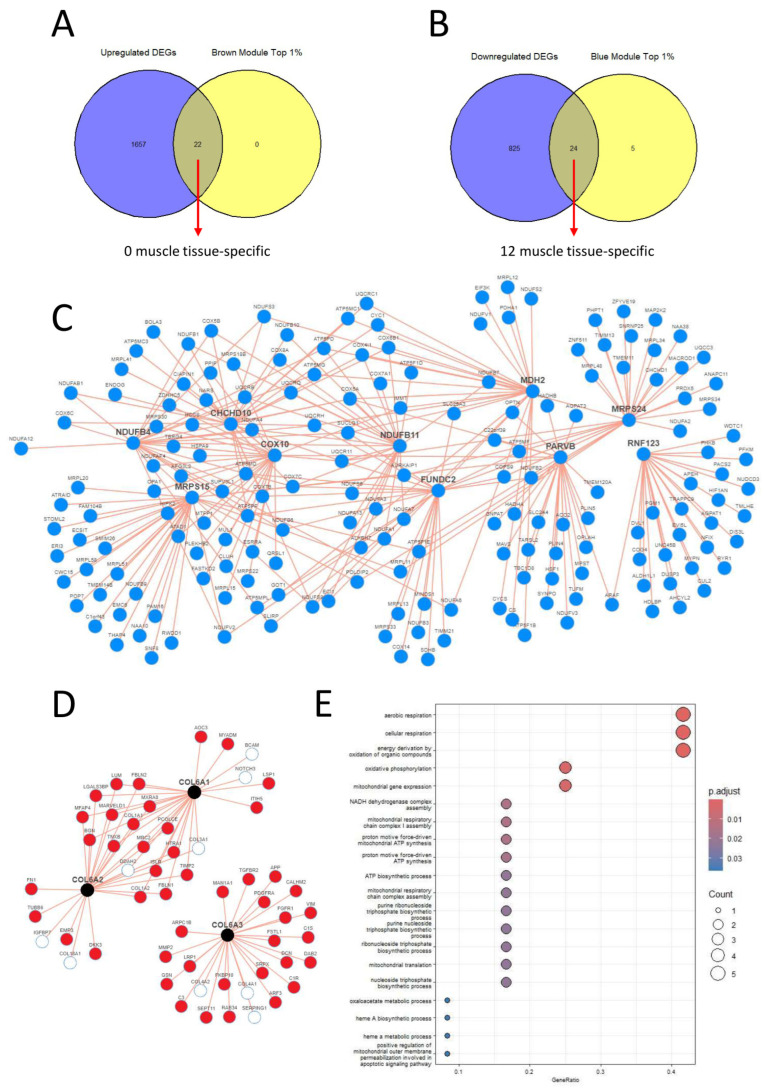
(**A**,**B**) We identified 22 genes shared between the top 1% upregulated modules and upregulated DEGs, and 24 genes common to the top 1% downregulated modules and downregulated DEGs. However, only 12 genes that exclusively have specific expression patterns for muscle tissue, based on the Human Protein Atlas database, were selected as targets. (**C**) Using the iNetModels database, a network graph of coexpression patterns for 10 selected genes was created. Node limits were set to 25 and red lines indicate positive correlations. (**D**) A coexpression map for *COL6A1*, *COL6A2*, and *COL6A3* was created using the iNetModels database. Red nodes show upregulated genes in our study, while white nodes are not differentially expressed. Red lines represent positive correlations between nodes. The node limit was set to 25, and the correlation parameter was set to “Both”. (**E**) Functional enrichment analysis results for the target genes are presented here, which were subsequently utilized in the Gene2drug tool.

**Figure 5 biomolecules-14-01376-f005:**
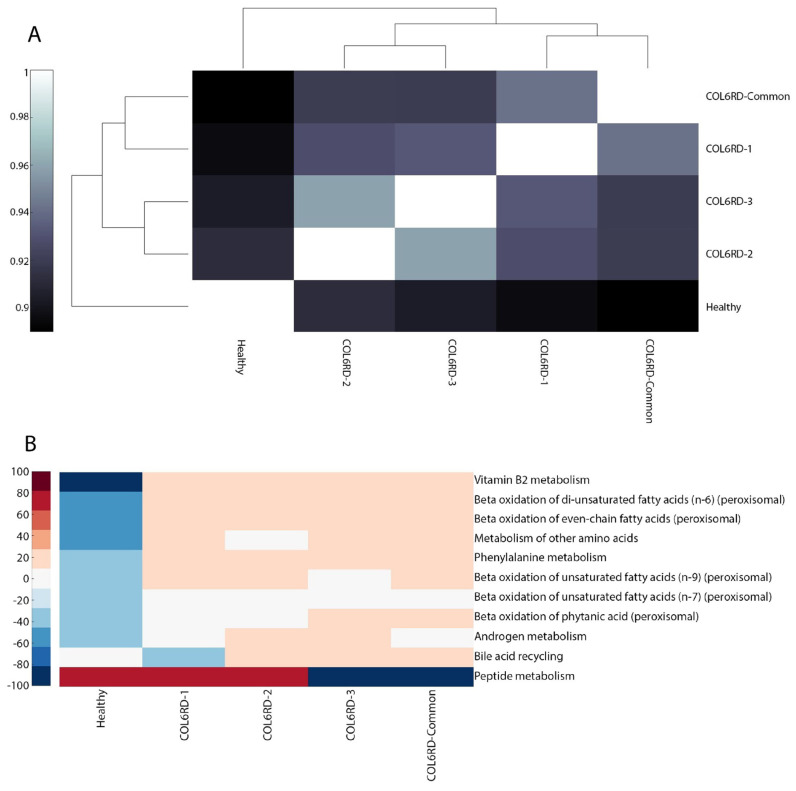
(**A**) A cluster gram displays Hamming distances between COL6RD and healthy GEMs, based on their reaction content and gene expressions. Notably, the Healthy GEM and COL6RD-Common model, generated from the average TPM of all COL6RD patients regardless of their histological grade, exhibit the most distinct pattern compared to others. (**B**) Another cluster gram highlights variations in subsystem coverage, focusing on those with at least a 25% variance in one or more GEMs. Redder tones signify higher coverage, while bluer tones denote lower coverage compared to other pathways. (**C**) A scatter plot depicts variations in the success or failure of metabolic tasks within genome-scale models. It is worth noting that genes with expression levels below 1 TPM are considered not expressed in the model while being built with the tINIT algorithm.

**Figure 6 biomolecules-14-01376-f006:**
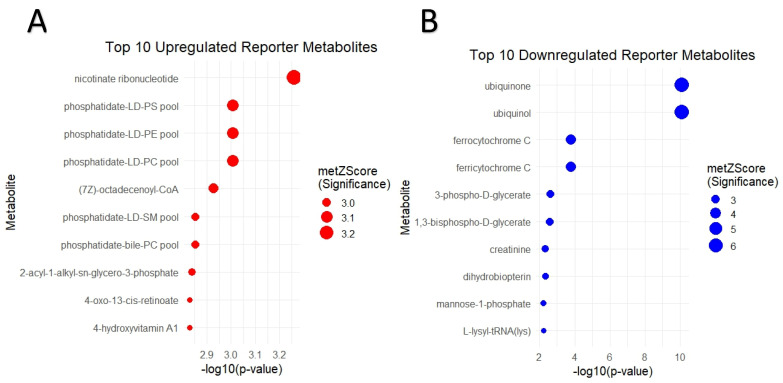
(**A**,**B**) Dot plots showcase the top ten upregulated and downregulated reporter metabolites, ranked by the lowest *p*-value. Upregulation is denoted by red dots, while downregulation is represented by blue dots. Each dot’s size corresponds to the metabolite’s Z score, with larger sizes indicating higher Z scores.

**Table 1 biomolecules-14-01376-t001:** List of downregulated target genes for COL6RD and their descriptions.

Gene Symbol	Description
CHCHD10	Coiled-Coil-Helix-Coiled-Coil-Helix Domain Containing 10 [49]
MRPS24	Mitochondrial Ribosomal Protein S24 [50]
TRIP10	Thyroid Hormone Receptor Interactor 10 [51]
RNF123	Ring Finger Protein 123 [52]
MRPS15	Mitochondrial Ribosomal Protein S15 [53]
NDUFB4	NADH: Ubiquinone Oxidoreductase Subunit B4 [54]
COX10	Cytochrome C Oxidase Assembly Factor Heme A: Farnesyltransferase COX10 [55]
FUNDC2	FUN14 Domain Containing 2 [56]
MDH2	Malate Dehydrogenase 2 [57]
RPL3L	Ribosomal Protein L3 Like [58]
NDUFB11	NADH: Ubiquinone Oxidoreductase Subunit B11 [59]
PARVB	Parvin Beta [60]

**Table 2 biomolecules-14-01376-t002:** Top 10 repurposed drugs for COL6RD and their enrichment scores, *p*-values and brief descriptions.

Drug Name	Enrichment Score	*p*-Value	Description
apigenin	0.578116	0.001314	a flavonoid found in various fruits, vegetables, and herbs [64]
flunarizine	0.564441	0.001850	a calcium channel blocker primarily used for migraine headaches [65]
deferoxamine	0.551187	0.002559	a chelating agent used to treat iron or aluminum toxicity [66]
luteolin	0.534912	0.003771	a type of flavonoid [67]
verteporfin	0.533962	0.003856	a benzoporphyrin derivative used as a photosensitizer [68]
ursodeoxycholic acid	0.513045	0.006232	a secondary bile acid with various therapeutic applications [69]
ioxaglic acid	0.507648	0.007032	an iodinated contrast medium used for X-ray imaging [70]
risperidone	0.500405	0.008252	an atypical antipsychotic drug used for schizophrenia [71]
fipexide	0.497805	0.008736	a psychoactive drug belonging to the piperazine chemical class [72]
naftifine	0.491820	0.009947	a broad-spectrum antifungal agent [73]

**Table 3 biomolecules-14-01376-t003:** Catalog of the top 10 reporter metabolites, including their regulatory status and concise explanations.

Metabolite	Regulation	Description
nicotinate ribonucleotide	Upregulated	a compound involved in nicotinate and nicotinamide metabolism [77]
phosphatidate-LD-PC pool	Upregulated	phosphatidate is an intermediate in the synthesis of various lipids [78]
phosphatidate-LD-PE pool	Upregulated	the pool of phosphatidate and phosphatidylethanolamine [79]
phosphatidate-LD-PS pool	Upregulated	the pool of phosphatidate and phosphatidylserine [80]
(7Z)-octadecenoyl-CoA	Upregulated	a compound involved in lipid metabolism [81]
ubiquinol	Downregulated	involved in cellular energy production and antioxidant protection [82]
ubiquinone	Downregulated	lays a crucial role in cellular energy production and antioxidant protection [83]
ferricytochrome C	Downregulated	a vital component of the electron transport chain in mitochondria [75]
ferrocytochrome C	Downregulated	a key component of the electron transport chain in mitochondria [75]
3-phospho-D-glycerate	Downregulated	an important metabolic intermediate in glycolysis [84]

## Data Availability

The RNA-seq data utilized in our research were obtained from publicly available data sources. They can be accessed through the following link: https://www.ncbi.nlm.nih.gov/geo/query/acc.cgi?acc=GSE103608, accessed on 3 June 2024.

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
