# Peer review of "Identifying Hub Genes and Metabolic Pathways in Collagen VI-Related Dystrophies: A Roadmap to Therapeutic Intervention"

_biomolecules, 2024, doi:10.3390/biom14111376_

Round 1
Reviewer 1 Report
Comments and Suggestions for Authors
To characterize the molecular mechanisms underlying COL6RD, the authors re-analyzed transcriptomic data from public sources related to COL6RD patients, identifying 2,528 differentially expressed genes. They then examined the co-expression patterns of these genes and identified nine modules. Focusing on two modules specific to muscle tissue expression, they pinpointed 12 genes as potential hub genes involved in the disease. To find potential pharmaceutical candidates, they performed gene ontology term analysis on these 12 target genes to identify drugs that might upregulate the relevant pathways. To detect context-specific metabolic changes, they employed genome-scale metabolic models (GEMs) tailored to COL6RD. Finally, they conducted reporter metabolite analysis and identified metabolites that were upregulated or downregulated in COL6RD.
If so, these findings would represent a novel methodology for uncovering critical genes and pathways associated with COL6RD, as well as identifying candidate drugs for therapeutic interventions.
However, there are some concerns about the presentation of the data and organization of the manuscript, and the interpretation of the results.
1. Organization of the manuscript
The manuscript is titled "Decoding Molecular Mechanisms of Collagen VI‐Related Muscular Dystrophies Using Genome-Scale Metabolic Modeling," yet the discussion of genome-scale metabolic modeling doesn't appear until the fifth section out of six in the results. This raises concerns about the organization and focus of the manuscript. The early sections primarily address other analyses, such as re-analysis of transcriptomic data, identification of DEGs, co-expression network analysis, and gene ontology analysis. Given that genome-scale metabolic modeling is featured prominently in the title, it seems disproportionate that it constitutes only a later part of the results. This organization may confuse readers about the primary focus of the study. To have a more logical flow that aligns with title, authoer can either reorganize the results structures or modify the title for accuracy.
- Structural flow: relevance of co-expression analysis in identifying hub genes
In the process of identifying hub genes as potential drug targets, the authors examined the co-expression patterns of genes and identified nine modules. By focusing on two modules specific to muscle tissue expression, they pinpointed 12 genes as potential hub genes involved in the disease. However, in lines 335-345, the authors state: "Additionally, we analyzed the co-expression patterns of the genes responsible for COL6RD by using the iNetModels database .... Figure 4D presents this co-expression comparison map and the correlations between the nodes." This additional analysis using the iNetModels database does not seem to provide further information relevant to identifying muscle tissue-specific expression. Instead, it appears to disrupt the flow of the manuscript. To improve the flow, authoer can either explain the added value using the iNetModels analysis or provide transitional statements.
- Consistency with previous findings
Are the results of this study consistent with previous findings regarding differentially expressed genes (DEGs) specific to COL6RD samples using the same data? The authors mention in the discussion (lines 421-424) that differences in their findings might be due to mapping to a more updated reference genome and using different RNA-seq tools. However, they do not quantify how these methodological differences impact the results. This raises questions about the overlap between the DEGs identified in this study and those from previous studies.
Specifically: a) How much alignment is there between the upregulated and downregulated genes in COL6RD samples relative to control samples between the two datasets?
b) Without quantifying the differences, it's unclear whether the discrepancies are significant or negligible. Are the DEGs identified in this study largely consistent with those identified previously, or do the methodological differences lead to substantially different sets of DEGs?
- Issues with figure presentation
a) All six figures in the manuscript are not properly presented—they appear compromised—making it difficult to see the details, especially when zoomed in or out.
b) In Figure 3D, the heatmap uses a red-blue color scale, where red indicates positive correlations and blue indicates negative correlations. However, the 9 modules are also colored in yellow, green-yellow, magenta, green, red, brown. Some of the module color labels (e.g., red, blue) overlap with the colors used in the heatmap's correlation scale. To avoid mistakenly associate the module color labels with the correlation values in the heatmap, distinct color palettes can be used.
Comments on the Quality of English LanguageSee above comments. By refining the manuscript's structure and flow of presentation, it will be better aligned with its title and results.
Author Response
Comments 1: To characterize the molecular mechanisms underlying COL6RD, the authors re-analyzed transcriptomic data from public sources related to COL6RD patients, identifying 2,528 differentially expressed genes. They then examined the co-expression patterns of these genes and identified nine modules. Focusing on two modules specific to muscle tissue expression, they pinpointed 12 genes as potential hub genes involved in the disease. To find potential pharmaceutical candidates, they performed gene ontology term analysis on these 12 target genes to identify drugs that might upregulate the relevant pathways. To detect context-specific metabolic changes, they employed genome-scale metabolic models (GEMs) tailored to COL6RD. Finally, they conducted reporter metabolite analysis and identified metabolites that were upregulated or downregulated in COL6RD.
If so, these findings would represent a novel methodology for uncovering critical genes and pathways associated with COL6RD, as well as identifying candidate drugs for therapeutic interventions.
However, there are some concerns about the presentation of the data and organization of the manuscript, and the interpretation of the results.
- Organization of the manuscript
The manuscript is titled "Decoding Molecular Mechanisms of Collagen VI‐Related Muscular Dystrophies Using Genome-Scale Metabolic Modeling," yet the discussion of genome-scale metabolic modeling doesn't appear until the fifth section out of six in the results. This raises concerns about the organization and focus of the manuscript. The early sections primarily address other analyses, such as re-analysis of transcriptomic data, identification of DEGs, co-expression network analysis, and gene ontology analysis. Given that genome-scale metabolic modeling is featured prominently in the title, it seems disproportionate that it constitutes only a later part of the results. This organization may confuse readers about the primary focus of the study. To have a more logical flow that aligns with title, authoer can either reorganize the results structures or modify the title for accuracy.
Response 1: Thank you for your insightful feedback. We have updated the article’s title to “Identifying Hub Genes and Metabolic Pathways in Collagen VI-Related Dystrophies: A Roadmap to Therapeutic Intervention.”
Comments 2: Structural flow: relevance of co-expression analysis in identifying hub genes
In the process of identifying hub genes as potential drug targets, the authors examined the co-expression patterns of genes and identified nine modules. By focusing on two modules specific to muscle tissue expression, they pinpointed 12 genes as potential hub genes involved in the disease. However, in lines 335-345, the authors state: "Additionally, we analyzed the co-expression patterns of the genes responsible for COL6RD by using the iNetModels database .... Figure 4D presents this co-expression comparison map and the correlations between the nodes." This additional analysis using the iNetModels database does not seem to provide further information relevant to identifying muscle tissue-specific expression. Instead, it appears to disrupt the flow of the manuscript. To improve the flow, authoer can either explain the added value using the iNetModels analysis or provide transitional statements.
Response 2: We appreciate your feedback. Our goal is to provide a more comprehensive overview of COL6RD, specifically focusing on its gene expression profile. To enhance clarity, we have changed the sentence at the beginning of the paragraph, as shown below:
“…To provide a more comprehensive overview of COL6RD’s gene expression profile, we also examined COL6A1, COL6A2 and COL6A3 gene co-expression patterns using the iNetmodels database…”
Comments 3: Consistency with previous findings
Are the results of this study consistent with previous findings regarding differentially expressed genes (DEGs) specific to COL6RD samples using the same data? The authors mention in the discussion (lines 421-424) that differences in their findings might be due to mapping to a more updated reference genome and using different RNA-seq tools. However, they do not quantify how these methodological differences impact the results. This raises questions about the overlap between the DEGs identified in this study and those from previous studies.
Specifically: a) How much alignment is there between the upregulated and downregulated genes in COL6RD samples relative to control samples between the two datasets?
- b) Without quantifying the differences, it's unclear whether the discrepancies are significant or negligible. Are the DEGs identified in this study largely consistent with those identified previously, or do the methodological differences lead to substantially different sets of DEGs?
Response 3: Thank you for bringing that point to our attention. To clarify, we outlined the similarities and differences between our study and the previous one, as detailed below:
“…Likewise, the initial research paper emphasized the increased activity of extracellular matrix and structural organization pathways while noting a decrease in mitochondrial pathways. However, there were notable differences. For instance, we identified 2,528 DEGs, whereas the original study reported only 248. Additionally, our gene ontology analysis revealed oxidative phosphorylation and mitochondrial energy metabolism as significant pathways for downregulated DEGs, contrasting with the original study's emphasis on fiber contraction and metabolic processes…”
Comments 4: Issues with figure presentation
- a) All six figures in the manuscript are not properly presented—they appear compromised—making it difficult to see the details, especially when zoomed in or out.
- b) In Figure 3D, the heatmap uses a red-blue color scale, where red indicates positive correlations and blue indicates negative correlations. However, the 9 modules are also colored in yellow, green-yellow, magenta, green, red, brown. Some of the module color labels (e.g., red, blue) overlap with the colors used in the heatmap's correlation scale. To avoid mistakenly associate the module color labels with the correlation values in the heatmap, distinct color palettes can be used.
Response 4: Thank you for your comment. We enhanced the resolution of all figures in the manuscript. Additionally, in Figure 3D, we modified the colour palette of the heatmap, with brown indicating positive correlation and purple indicating negative correlation.
Reviewer 2 Report
Comments and Suggestions for Authors
This article "Deciphering the Molecular Mechanisms of Collagen VI-Associated Muscular Dystrophies Using Genome-Scale Metabolic Modeling" is interesting and deserves to be published in Biomolecules. The authors identified 12 genes (CHCHD10, MRPS24, TRIP10, RNF123, MRPS15, NDUFB4, COX10, FUNDC2, 24 MDH2, RPL3L, NDUFB11, PARVB) as potential hub genes involved in the disease. The authors used a drug repurposing strategy to identify pharmaceutical candidates that could potentially modulate these genes and be effective in treatment.
All figures are of low quality.
There is an error in the number of patients in the figure. It should be 14 for healthy subjects.
Figure 3D. Significance levels are indicated by asterisks (*p<0.05). What is the p-value for * and **?
Figure 5A. If "0" is "different" and "1" is "similar", then it is clear from the figure that the greatest similarity is between COL6RD-3 and COL6RD-2. And the text (line 379) says that "...COL6RD-1 and COL6RD-Common showed the greatest similarity..."
Author Response
Comments 1: This article "Deciphering the Molecular Mechanisms of Collagen VI-Associated Muscular Dystrophies Using Genome-Scale Metabolic Modeling" is interesting and deserves to be published in Biomolecules. The authors identified 12 genes (CHCHD10, MRPS24, TRIP10, RNF123, MRPS15, NDUFB4, COX10, FUNDC2, 24 MDH2, RPL3L, NDUFB11, PARVB) as potential hub genes involved in the disease. The authors used a drug repurposing strategy to identify pharmaceutical candidates that could potentially modulate these genes and be effective in treatment.
All figures are of low quality.
Response 1: Thank you for your helpful feedback. We have improved the resolution of all figures embedded in the manuscript file.
Comments 2: There is an error in the number of patients in the figure. It should be 14 for healthy subjects.
Response 2: Thank you for bringing that mistake to our attention. The figure has been corrected.
Comments 3: Figure 3D. Significance levels are indicated by asterisks (*p<0.05). What is the p-value for * and **?
Response 3: A single asterisk (*) denotes a p-value below 0.05, a double asterisk (**) denotes a p-value below 0.01, and a triple asterisk (***) denotes a p-value below 0.001. This information has been added to the figure legends.
Comments 4: Figure 5A. If "0" is "different" and "1" is "similar", then it is clear from the figure that the greatest similarity is between COL6RD-3 and COL6RD-2. And the text (line 379) says that "...COL6RD-1 and COL6RD-Common showed the greatest similarity..."
Response 4: Thank you for your comment. We have revised the sentence as follows:
“…Furthermore, COL6RD-3 and COL6RD-2 displayed the highest similarity, with a Hamming distance score of 0.96. This was followed by the COL6RD-1 and COL6RD-Common models…”
Reviewer 3 Report
Comments and Suggestions for Authors
The article studies COL6RD using systems biology, identifying hub genes consisting of 12 potential drug targets. However, I see a number of facts unacceptable for a scientific article: many of the images are so small that it is impossible to see them. It is not clear what they mean. When zooming in, the quality is lost. In Table 1, the last column is the same for all values. Why do it then? The conclusion is very short. The level of borrowing from someone else's text is very high. I read the work several times and did not understand anything.
Author Response
Comments 1: The article studies COL6RD using systems biology, identifying hub genes consisting of 12 potential drug targets. However, I see a number of facts unacceptable for a scientific article: many of the images are so small that it is impossible to see them. It is not clear what they mean. When zooming in, the quality is lost.
Response 1: Thank you for your helpful feedback. We have improved the resolution of all figures in the manuscript file.
Comments 2: In Table 1, the last column is the same for all values. Why do it then?
Response 2: Thank you for your comment. The last column has been removed from the table.
Comments 3: The conclusion is very short.
Response 3: We appreciate your feedback on our work. The conclusion section has been expanded to provide a more detailed summary of the study's findings, as shown below:
"In conclusion, our study on COL6RD using systems biology has offered important insights into the disease’s pathophysiology. By integrating differential expression analysis with WGCNA, we pinpointed hub genes and identified 12 potential drug targets. Then, we applied the pathway-based drug repositioning strategy to discover pharmacotherapy candidates aimed at modulating these genes. Additionally, we explored metabolic differences between healthy individuals and COL6RD patients by reconstructing and comparing context-specific genome-scale metabolic models. This was followed by reporter metabolite analysis, which helped identify key metabolites that may be valuable for early detection and monitoring. Briefly, our study identified hub genes, drug targets, key metabolites and pathways linked to COL6RD, and promising drug candidates for improving patient care. It provides a foundation for future metabolomic and clinical research to validate these findings."
Comments 4: The level of borrowing from someone else's text is very high. I read the work several times and did not understand anything.
Response 4: We are grateful to get your valuable feedback. We have revised the manuscript to lower the similarity index and improve clarity.
Reviewer 4 Report
Comments and Suggestions for Authors
The present paper approaches a crucial topic in muscular dystrophies, that is the search for effective treatment strategies that are able to mitigate the phenotype. They focused on Collagen VI-ralated muscular dystrophies, disorders associated with mutations on genes coding for Collagen VI chains. Indeed Bethlem Myopathy and Ullrich Congenital Muscular Dystrophies are very rare, but nevertheless they are strongly invalidating and in the worse cases also life threatening. For this reason every approach that is able to disclosure new possible therapeutic targets or even better identify repurpose already known drugs is of great importance. In this paper, taking advantages of transcriptomics data from previous studies on patients’ muscle tissues, the authors applied many different bioinformatics tolls, the more recent ones, in order to identify differentially expressed genes, coexpression network and potential drug candidates. Moreover, they defined gene targets and confirmed they expression in muscle tissues before investigating the metabolic proteins that were deregulated in the patients samples compared to control ones. The main results are the identification of 10 repurposed drugs, most of which are of antioxidant and anti-inflammatory action, and the highlighting of lipid metabolic deregulation.
The paper is clearly written and the data are effectively presented. The discussion highlights the limitations of the study and the need of further experimental evidences before generalizing the results.
Here find some comments.
· First of all, it would beneficial to prepare the tables in the way that each drugs or metabolites is divided from the following one by a line, otherwise it is quite difficult to retrieve all the information per each one. Add references whenever possible.
· There is no mention of the analyses on metabolic deregulation that were published in the mouse model of Bethlem myopathy (see Capitanio et al, 2016).
Other comments:
· Line 115: “we solely chose protein-coding genes for our analysis”. What about non-coding RNAs? This will help to better understand gene expression regulation. Please comment
· Line 179: “we selected genes with certain RNA expression levels in muscle tissue (tissue-enriched or tissue-enhanced) as our targets”. And what about the connective tissue and adipose tissue? As they are both involved in the pathology. Please comment.
· Line 222: why did they exclude these metabolites from the reporter metabolite analysis since they are involved in numerous biochemical reactions? Could the authors comment on it? Isn’t this a limitation for early detection/diagnosis and monitoring?
· Line 255: they should stress that this is a sort of validation of the paper they cite.
· Line 289: the modules tested are 13, not 12
· Line 339-340: The reference #50 is not really appropriate. Better cite Gara et al., 2008 J Biol Chem
· 409-410: there is a missing reference
· Figure 1: the number of samples is inverted: COL6RD are 22 not 14, Healthy are 14 not 22.
Author Response
Comments 1: The present paper approaches a crucial topic in muscular dystrophies, that is the search for effective treatment strategies that are able to mitigate the phenotype. They focused on Collagen VI-ralated muscular dystrophies, disorders associated with mutations on genes coding for Collagen VI chains. Indeed Bethlem Myopathy and Ullrich Congenital Muscular Dystrophies are very rare, but nevertheless they are strongly invalidating and in the worse cases also life threatening. For this reason every approach that is able to disclosure new possible therapeutic targets or even better identify repurpose already known drugs is of great importance. In this paper, taking advantages of transcriptomics data from previous studies on patients’ muscle tissues, the authors applied many different bioinformatics tolls, the more recent ones, in order to identify differentially expressed genes, coexpression network and potential drug candidates. Moreover, they defined gene targets and confirmed they expression in muscle tissues before investigating the metabolic proteins that were deregulated in the patients samples compared to control ones. The main results are the identification of 10 repurposed drugs, most of which are of antioxidant and anti-inflammatory action, and the highlighting of lipid metabolic deregulation.
The paper is clearly written and the data are effectively presented. The discussion highlights the limitations of the study and the need of further experimental evidences before generalizing the results.
Here find some comments.
First of all, it would beneficial to prepare the tables in the way that each drugs or metabolites is divided from the following one by a line, otherwise it is quite difficult to retrieve all the information per each one. Add references whenever possible.
Response 1: Thank you for your feedback. We have added lines to the table for better readability and included references.
Comments 2: There is no mention of the analyses on metabolic deregulation that were published in the mouse model of Bethlem myopathy (see Capitanio et al, 2016).
Response 2: Thank you for pointed out that mistake. We have changed reference to “Irwin, William A., et al. "Mitochondrial dysfunction and apoptosis in myopathic mice with collagen VI deficiency." Nature genetics 35.4 (2003): 367-371.”
Comments 3: Line 115: “we solely chose protein-coding genes for our analysis”. What about non-coding RNAs? This will help to better understand gene expression regulation. Please comment
Response 3: A key aspect of our study is identifying protein targets for COL6RD and modulating them with drugs to alter their activity. While non-coding RNAs aid in understanding gene regulation, they are beyond our current research scope. Therefore, we solely focus on protein coding genes. To clarify, we have rephrased the sentence as follows:
“We selected only protein-coding genes annotated in the BioMart data mining tool because a core objective of our study is to identify protein targets and modulate their activity with drugs.”
Comments 4: Line 179: “We selected genes with certain RNA expression levels in muscle tissue (tissue-enriched or tissue-enhanced) as our targets”. And what about the connective tissue and adipose tissue? As they are both involved in the pathology. Please comment.
Response 4: Thank you for your comment. While COL6RD impacts connective tissue, joints, skin, spine, and adipose tissue, its strongest effect is on muscle tissue. This can be seen in the results of the gene ontology analysis, which shows a significant downregulation in mitochondrial energy metabolism. Its common symptoms include progressive muscle weakness, muscle atrophy, delayed motor milestones, and loss of independent ambulation in severe patients. Given COL6RD’s primary effect on muscles, we hypothesized that targeting muscle-specific genes could minimize systemic side effects and improve therapy success when manipulating these genes with drugs.
Comments 5: Line 222: why did they exclude these metabolites from the reporter metabolite analysis since they are involved in numerous biochemical reactions? Could the authors comment on it? Isn’t this a limitation for early detection/diagnosis and monitoring?
Response 5: Thank you for your feedback. Currency metabolites, which are involved in many biochemical reactions, are often excluded from metabolic network analysis to improve clarity and focus. Their widespread presence can obscure specific changes, add noise, and make it harder to detect meaningful patterns. By removing these common metabolites, the analysis can better highlight relevant pathway-specific changes and improve statistical power by reducing unnecessary tests. We have included our comments on that topic in the manuscript, as shown below:
“Before running the RMA, 20 currency metabolites (“H2O”, “CO2”, “O2”, “H+”, “HCO3−”, “Na+”, “CoA”, “Pi”, “PPi”, “AMP”, “ADP”, “ATP”, “NAD+”, “NADH”, “NADP+”, “NADPH”, “PAP”, “PAPS”, “FAD” and “FADH2”) were excluded from the model because their widespread involvement in reactions can obscure specific changes, add noise, and reduce clarity. Their removal highlighted relevant pathway-specific changes and improved statistical power.”
Comments 6: Line 255: they should stress that this is a sort of validation of the paper they cite.
Response 6: Thank you for your feedback. We referenced the original paper to highlight that it serves as a form of validation.
“…Likewise, the initial research paper emphasized the increased activity of extracellular matrix and structural organization pathways while noting a decrease in mitochondrial pathways…”
Comments 7: Line 289: the modules tested are 13, not 12
Response 7: We are grateful for your comment. The genes that could not be assigned to any specific module were placed in the grey module. However, this module is not considered meaningful in WGCNA, which is why we mentioned a total of 12 modules.
Comments 8: Line 339-340: The reference #50 is not really appropriate. Better cite Gara et al., 2008 J Biol Chem
Response 8: Thank you for highlighting that error. We have now updated the reference to: "Gara, S. K., Grumati, P., Urciuolo, A., Bonaldo, P., Kobbe, B., Koch, M., Paulsson, M., & Wagener, R. (2008). Three novel collagen VI chains with high homology to the α3 chain. Journal of Biological Chemistry, 283(16), 10658–10670."
Comments 9: 409-410: there is a missing reference
Response 9: We have added related references to the section.
Comments 10: Figure 1: the number of samples is inverted: COL6RD are 22 not 14, Healthy are 14 not 22.
Response 10: Thank you for bringing that mistake to our attention. The figure has been corrected.
Round 2
Reviewer 3 Report
Comments and Suggestions for Authors
Dear authors, thank you for listening to my opinion and enlarging the pictures. Now they can be examined. It remains only to solve minor issues with the design, for example, links not in round brackets, but in square ones, and the article can be published.